# Evaluating WorldClim Version 1 (1961–1990) as the Baseline for Sustainable Use of Forest and Environmental Resources in a Changing Climate

**Maurizio Marchi [1],\*** , **Iztok Sinjur [2]**, **Michele Bozzano [3]** and **Marjana Westergren [2]**

[1] CREA—Research Centre for Forestry and Wood, I-52100 Arezzo, Italy
[2] Slovenian Forestry Institute, Vecna pot 2, 1000 Ljubljana, Slovenia; iztok.sinjur@gozdis.si (I.S.); marjana.westergren@gozdis.si (M.W.)
[3] European Forest Institute, 53113 Bonn, Germany; michele.bozzano@efi.int
\* Correspondence: maurizio.marchi@crea.gov.it; Tel.: +39-0575-353021; Fax: +39-0575-353490

**Abstract:** WorldClim version 1 is a high-resolution, global climate gridded dataset covering 1961–1990; a "normal" climate. It has been widely used for ecological studies thanks to its free availability and global coverage. This study aims to evaluate the quality of WorldClim data by quantifying any discrepancies by comparison with an independent dataset of measured temperature and precipitation records across Europe. BIO1 (mean annual temperature, MAT) and BIO12 (mean total annual precipitation, MAP) were used as proxies to evaluate the spatial accuracy of the WorldClim grids. While good representativeness was detected for MAT, the study demonstrated a bias with respect to MAP. The average difference between WorldClim predictions and climate observations was around +0.2 °C for MAT and −48.7 mm for MAP, with large variability. The regression analysis revealed a good correlation and adequate proportion of explained variance for MAT (adjusted $R^2$ = 0.856) but results for MAP were poor, with just 64% of the variance explained (adjusted $R^2$ = 0.642). Moreover no spatial structure was found across Europe, nor any statistical relationship with elevation, latitude, or longitude, the environmental predictors used to generate climate surfaces. A detectable spatial autocorrelation was only detectable for the two most thoroughly sampled countries (Germany and Sweden). Although further adjustments might be evaluated by means of geostatistical methods (i.e., kriging), the huge environmental variability of the European environment deeply stressed the WorldClim database. Overall, these results show the importance of an adequate spatial structure of meteorological stations as fundamental to improve the reliability of climate surfaces and derived products of the research (i.e., statistical models, future projections).

**Keywords:** spatial analysis; 1961–1990 normal period; spatial interpolation; geostatistics; ecological mathematics

## 1. Introduction

Easy access to standardized climate data with global coverage is paramount for the advancement of many ecological studies and to understand future ecosystem services provided by forest systems [1–3] and productive lands in agriculture [4,5]. One of the main aims for researchers dealing with environmental resources has become to forecast possible impacts of climate change on organisms and to evaluate possible mitigation [6–9]. In the past few decades, many conservation strategies have been suggested in order to maintain human well-being and ensure an adequate level of welfare [10] from (relatively) simple management strategies [4,11], including "assisted migration" [12–14], a controversial protocol that includes translocating more adapted or resilient genotypes for conservation or to improve the resilience of ecosystems. Such efforts are often driven by statistical models [14–17] and management

simulators [18,19], with both genetic variation and phenotypic plasticity included in the statistical models as covariates [20–22]. However, despite modelling efforts, such studies always require absolutely reliable climate data to be used as both baseline (e.g., 30-year average climate data) and for future predictions. Furthermore, while the uncertainty around GCMs and future trajectories is well known [23–25], information on current ecological limits of forest tree species has also been questioned [26]. In this context, the interest of researchers in gridded climate datasets has grown strongly.

The interpolation method, the spatial resolution and the coverage are the three main features that researchers use to select the most suitable datasets for their research [27–30]. The first release of the WorldClim dataset [31] is probably the most famous gridded climate dataset, widely used for ecological studies and freely available from (www.worldclim.org). Thanks to its high resolution (30 arc-second in the WGS84 reference system and approximately 1 km at the equator), global coverage, and availability, it has been used and cited more than 5200 times since publication [31]. The dataset is suitable for basic and applied studies in ecology, including forestry and ecological modeling [32–34], as well as to construct related datasets such as bio-geographical zones or environmental stratifications [35]. One of the main products of this database is "version 1", representative of the 1961–1990 climate normal period for the whole globe, including Antarctica. This version 1 dataset was generated by interpolating weather station data with the ANUSPLIN software (version 4.3) using latitude, longitude, and elevation as independent variables. The software implements a thin-plate smoothing spline procedure, using every station as a data point. A second-order spline function was fitted by the Authors using the above three variables, which produced the lowest overall cross-validation errors [31]. Considering the ANUSPLIN program creates a continuous surface projection, the LAPGRD program was used to create a global grid of climate surfaces with 30 arc-seconds horizontal and vertical resolution commonly referred to as 1 km$^2$ resolution. Raster maps for monthly precipitation amount and mean, maximum, and minimum air temperature were then provided. Raw data came from weather stations retrieved from various databases including GHCN, WMO climatological normals, FAOCLIM 2.0, CIAT, and regional databases and, where possible, restricted to the period 1950–2000. Quality control measures were taken to remove duplicate records, giving precedence to the GHCN database. After the quality control check and cleaning, the database consisted of precipitation records from 47,554 locations and mean air temperature from 24,542 locations [31]. Then elevation bias in weather stations was related to latitude and presence of mountain ranges. However, local records from many European countries were not easily accessible and WorldClim climate surfaces for Europe were constructed using 1263 records for air temperature and 2116 for precipitation.

WorldClim version 1 has recently been acknowledged to be representative of the 1961–1990 climate normal period. This time-slice has been widely used as the pre-industrial climate in many papers about the potential impact of climate change on ecosystems [1,3,26,28,36,37] and other ecological fields. Nevertheless, given the detailed description provided by the Authors in their paper, the question remains whether the quality of the WorldClim climate surfaces as a proxy of the climate baseline is adequate in complex environments such as, for instance, the European environment.

The present study aims to assess and quantify the reliability of WorldClim climate raster maps for Europe. We compared WorldClim with observed average values for mean annual temperature and total annual precipitation for the period 1961–1990. Data were retrieved for the whole of Europe building an independent dataset with data from many meteorological services. Then statistical analysis was run in order to evaluate the reliability of this dataset across the study area.

## 2. Materials and Methods

### 2.1. Construction and Description of the Database Used for Comparison

To investigate WorldClim's reliability in predicting baseline climate conditions we compiled an independent climate dataset by collecting data from weather services across Europe which were already freely available or delivered upon request (Table 1). All data were specifically requested or

downloaded as monthly averaged values over the 30-year normal period (1961–1990). Local monthly air temperature averages (MAT) and precipitation sums (MAP) were aggregated to calculate annual values. In total we retrieved data from 6659 meteostations across Europe, with 1759 records for temperature and 6526 records for precipitation (Figure 1). Most of the records were retrieved for Germany and Sweden with 4825 and 1391 meteostations, respectively, while for some countries, records were much fewer (e.g., Spain, France, Italy) or totally absent (e.g., Serbia, Poland, Romania).

Nevertheless, even if not equally distributed geographically, neither balanced concerning the ecological regions of Europe, we considered the distribution of the collected data as adequate for the purpose. Despite the lack of uniform coverage of both geography and ecological regions, we considered the data collected to be adequate for subsequent analysis.

Moreover we tested the random distribution of MAT and MAP with the randtest package of the R statistical language [38]. The database was carefully checked and cleaned to remove entries with missing data and to geo-reference each record. Very few points (112), corresponding to less than 1% of all the records, lay outside country borders or land masses due to coordinate uncertainties, which reflects the high-quality of the new database. Such records were removed completely from the database in order to avoid any influences on the calculations.

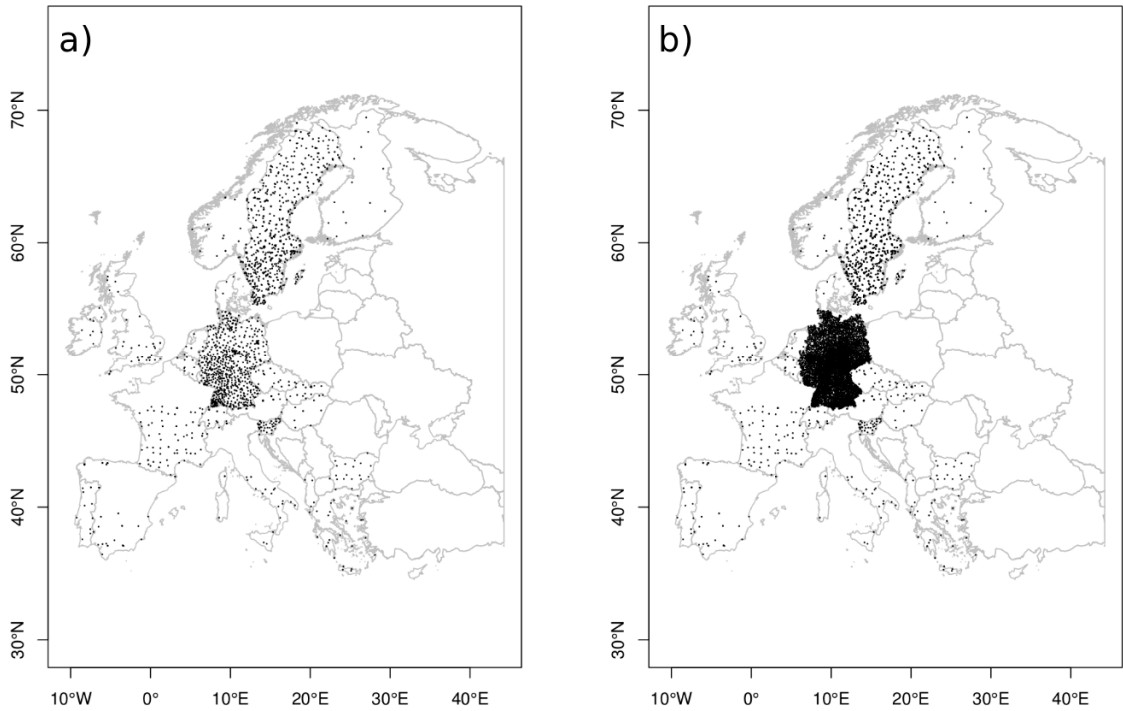

**Figure 1.** Spatial distribution of the compiled dataset for: (**a**) temperature records and (**b**) precipitation records. Each dot represents a meteorological station. The darker the area, the more data were retrieved.

**Table 1.** Structure of the compiled database.

| Country | Total Meteostations | MAT Records | MAP Records | Data Source |
|---|---|---|---|---|
| Albania | 3 | 3 | 3 | [39] |
| Austria | 23 | 21 | 23 | |
| Belgium | 9 | 9 | 9 | |
| Bulgaria | 18 | 18 | 18 | |
| Croatia | 1 | 1 | 1 | |
| Czech | 20 | 19 | 20 | |
| Denmark | 4 | 4 | 4 | |
| Finland | 18 | 17 | 18 | |
| France | 76 | 75 | 76 | |
| Germany | 4825 | 719 | 4733 | [40] |
| Greece | 26 | 25 | 26 | [41] |
| Hungary | 9 | 9 | 9 | [39] |
| Ireland | 6 | 6 | 6 | |
| Italy | 30 | 30 | 30 | |
| North Macedonia | 1 | 1 | 1 | |
| Montenegro | 1 | 1 | 1 | |
| Netherlands | 5 | 5 | 5 | |
| Norway | 18 | 18 | 18 | |
| Portugal | 18 | 18 | 18 | |
| Slovakia | 14 | 14 | 14 | |
| Slovenia | 42 | 42 | 42 | [42] |
| Spain | 51 | 51 | 51 | [39] |
| Sweden | 1391 | 604 | 1351 | [43] |
| Switzerland | 12 | 11 | 12 | [39] |
| United Kingdom | 38 | 38 | 37 | |
| *TOTAL* | *6659* | *1759* | *6526* | |
| *MEAN* | *266* | *70* | *261* | |
| *ST. DEV* | *988.64* | *179.54* | *969.08* | |

## 2.2. Comparisons and Statistical Procedures

The BIO1 (mean annual air temperature) and BIO12 (mean total annual precipitation) variables of WorldClim were used as proxies to evaluate the spatial accuracy of raster surfaces. The strata were first downloaded from the official WorldClim web portal. Then, using an overlay function, the corresponding values of the two climate variables were extracted for each meteorological station in our database. A linear regression analysis was then applied to analyze the relationships between the predicted WorldClim value and the observed value in our dataset. The adjusted $R^2$ was used to measure the amount of environmental variability expressed by WorldClim. Then the difference between the WorldClim value and the observed value (30-years normal value from our database) was calculated for each location of our database. To avoid confusion and mathematical balancing between positive and negative values, which might seriously affect the analysis, both the raw discrepancy (BIAS) and its absolute value (ABIAS) were calculated. To study possible trends across the data, we looked at the relationships between BIAS and the predictors used by the authors

of WorldClim during the spatial interpolation process (i.e., latitude, longitude, elevation). Then, we retrieved the complete database of meteorological stations used by the WorldClim authors from www.arcgis.com/home/item.html?id=7644c6e78c1644b4bde2edfc44787520) and clipped to the European environment (Table 2).

We calculated the average distance of each meteorological station in our database from the geographically closest five stations in the WorldClim dataset. We expected a smaller difference where WorldClim stations were denser. Finally, the spatial autocorrelation of BIAS was evaluated using geostatistical analysis implemented in R using the gstat package [44] and modelling the semivariance of BIAS as a function of the spatial distance between records.

The whole structure of the data collection and analysis procedure is graphically reported on Figure 2.

Table 2. Number of meteorological stations per country used by Hijmans et al. [31] in Europe.

| Country | Temperature | Precipitation | Country | Temperature | Precipitation |
|---|---|---|---|---|---|
| Albania | 0 | 7 | Latvia | 3 | 9 |
| Andorra | 0 | 0 | Liechtenstein | 0 | 0 |
| Armenia | 2 | 2 | Lithuania | 16 | 19 |
| Austria | 3 | 25 | Luxembourg | 1 | 6 |
| Belarus | 8 | 22 | North Macedonia | 7 | 7 |
| Belgium | 3 | 18 | Malta | 1 | 3 |
| Bosnia and Herz. | 7 | 10 | Moldova | 2 | 3 |
| Bulgaria | 4 | 15 | Monaco | 0 | 0 |
| Croatia | 13 | 13 | Montenegro | 5 | 2 |
| Czech Republic | 7 | 16 | Netherlands | 7 | 10 |
| Denmark | 19 | 41 | Norway | 8 | 54 |
| Estonia | 3 | 12 | Poland | 18 | 63 |
| Faeroe Islands | 1 | 1 | Portugal | 16 | 18 |
| Finland | 19 | 32 | Romania | 11 | 28 |
| France | 82 | 107 | Russia | 44 | 124 |
| Georgia | 1 | 20 | San Marino | 0 | 0 |
| Germany | 89 | 116 | Serbia | 23 | 12 |
| Gibraltar | 0 | 1 | Slovakia | 3 | 10 |
| Greece | 26 | 48 | Slovenia | 6 | 2 |
| Guernsey | 0 | 0 | Spain | 60 | 117 |
| Hungary | 8 | 20 | Sweden | 16 | 60 |
| Ireland | 16 | 51 | Switzerland | 8 | 20 |
| Isle of Man | 0 | 1 | Turkey | 513 | 548 |
| Italy | 133 | 151 | Ukraine | 22 | 81 |
| Jersey | 0 | 3 | UK | 29 | 188 |
| **Summary statistics** | **Temperature records** | | | **Precipitation records** | |
| TOTAL | 1263 | | | 2116 | |
| MEAN | 25 | | | 42 | |
| SD | 74.86 | | | 84.89 | |

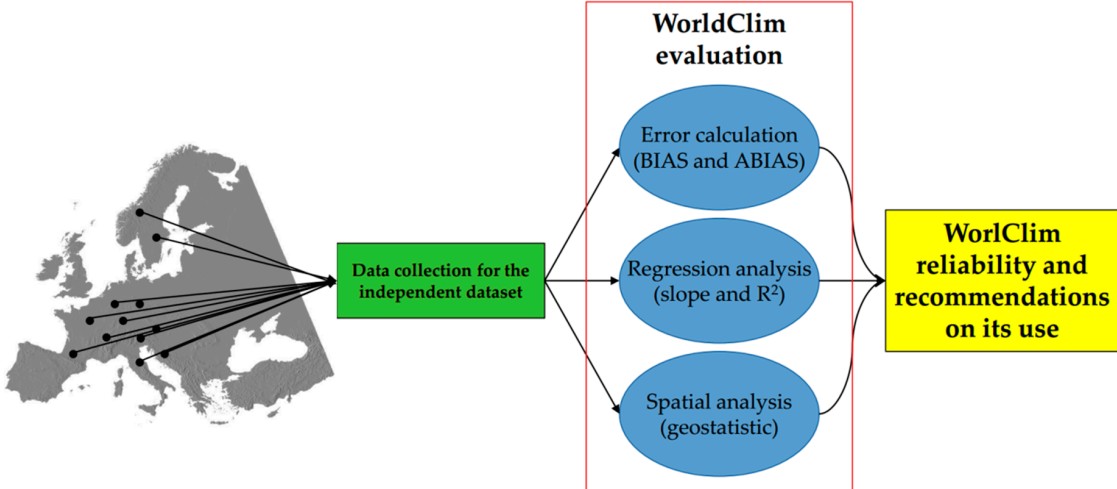

**Figure 2.** Flowchart of the data collection and statistical analysis we made to test the reliability of WorlClim version 1 data.

## 3. Results

The compiled database included 25 European countries, albeit with an unbalanced distribution. Overall, an average of 266 records per country (both MAT and MAP) was included in the database. However, the difference among countries was huge, with a standard deviation of ± 988.64 records per country. This large standard deviation was caused by the disproportionate number of records for Sweden and Germany. Temperature (MAT) values ranged from −5.8 °C to 21.2 °C, while precipitation (MAP) was between 104.8 mm and 3318 mm. The mean difference between the interpolated WorldClim values and the observed values was 0.22 °C for temperature and −48.7mm for precipitation (Table 3), with a high coefficient of variation (6.82 for MAT and 3.40 for MAP). BIAS ranged between −10.6 °C and 13.2 °C for MAT and between −1578.1 mm and 950.8 mm for MAP. Mean ABIAS was 0.76 for MAT and 98.56 for MAP.

Results of the regression analysis for MAT and MAP are shown in Figure 3. Residuals of linear models were randomly distributed for both of the analyzed variables and were highly significant ($p < 2.2 \times 10^{-16}$). Concerning MAT, the good correlation and adequate proportion of explained variance point to a low discrepancy between the two datasets; WorldClim explained 86% of the variance (adjusted $R^2 = 0.856$) with a residual random standard error of 1.50°C, intercept of −0.202 °C and slope almost equal to 1 (0.996). The regression line and the expected regression line for a perfect match between the two datasets almost overlapped. For MAP, 64% (adjusted $R^2 = 0.642$) of the variance of the precipitation dataset was explained by a linear regression model, with a residual standard error of 159.6 mm. The match between the two regression lines was considerably low (Figure 3, right) with the slope of the regression coefficient higher than 1. WorldClim was characterized by higher values than observed under 500 mm precipitation and lower values above this threshold. As overall, a general overestimation of MAP values was detected in dry areas (<500 mm) with an underestimation in the remaining zones.

**Table 3.** Difference between local data and WorldClim's surfaces.

| Variable | AVR | SD | CV | MAX | MIN | ABSAVR |
|----------|-----|-----|-----|-----|-----|--------|
| MAT [°C] | 0.22 | 1.50 | 6.82 | −10.62 | 13.21 | 0.76 |
| MAP [mm] | −48.70 | 165.35 | 3.40 | −1578.10 | 950.80 | 98.56 |

AVR = average value; SD = standard deviation; CV = coefficient of variation; MAX = maximum difference; MIN = minimum difference; ABSAVR = average of absolute values.

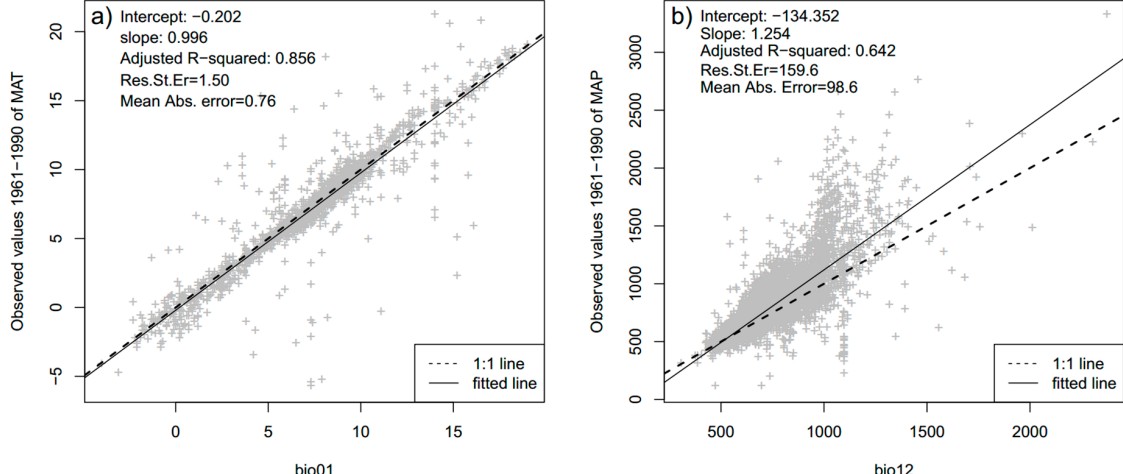

**Figure 3.** Results of the regression analysis for: (**a**) temperature represented by Bio1 variable of WorldClim database on x-axis and (**b**) precipitation represented by Bio12 variable of WorldClim database versus observed values on the y-axis. Regression coefficients at top-left of each figure. The 1:1 line of a perfect match shown dashed.

No relationship was found between the modelling error (ER) detected for MAT and MAP and the environmental predictors used for spatial interpolation across Europe. Modelled linear regressions explained less than 5% of variation, with one exception (Table 4). This lack of correlation can also be observed in Figure 4 where ABIAS is plotted against the average spatial distance of the "observed" meteorological station from the five WorldClim stations.

**Table 4.** Linear regression parameters when modelling error (ER) and the environmental predictors. Each predictor was tested separately (ADF5NM=Average distance from the five nearest meteorological stations).

| Variable | Predictor | Intercept | Slope | Explained Variance | *p*-Value |
|---|---|---|---|---|---|
| MAT | Latitude | 0.29 | 0.000000 | 0.56% | 0.00092 |
| | Longitude | −0.34 | 0.000000 | 1.20% | 0.00000 |
| | Elevation | 0.54 | −0.001025 | 4.95% | 0.00000 |
| | ADF5NM | 0.09 | 0.000002 | 0.18% | 0.04138 |
| MAP | Latitude | −45.16 | 0.000014 | 0.05% | 0.04492 |
| | Longitude | −202.28 | 0.000061 | 4.33% | 0.00000 |
| | Elevation | 8.15 | −0.206690 | 10.26% | 0.00000 |
| | ADF5NM | −107.02 | 0.001059 | 1.61% | 0.00000 |

The spatial distribution of BIAS in the two most represented countries is shown in Figure 5 for the two investigated variables. Spatial aggregation is especially evident in Sweden, where most of the "large dots" are clustered in the south of the country. For Sweden and Germany, variograms of the MAP variable were fitted by means of an exponential variogram model and revealed a clear spatial autocorrelation, especially for Germany (Figure 6).

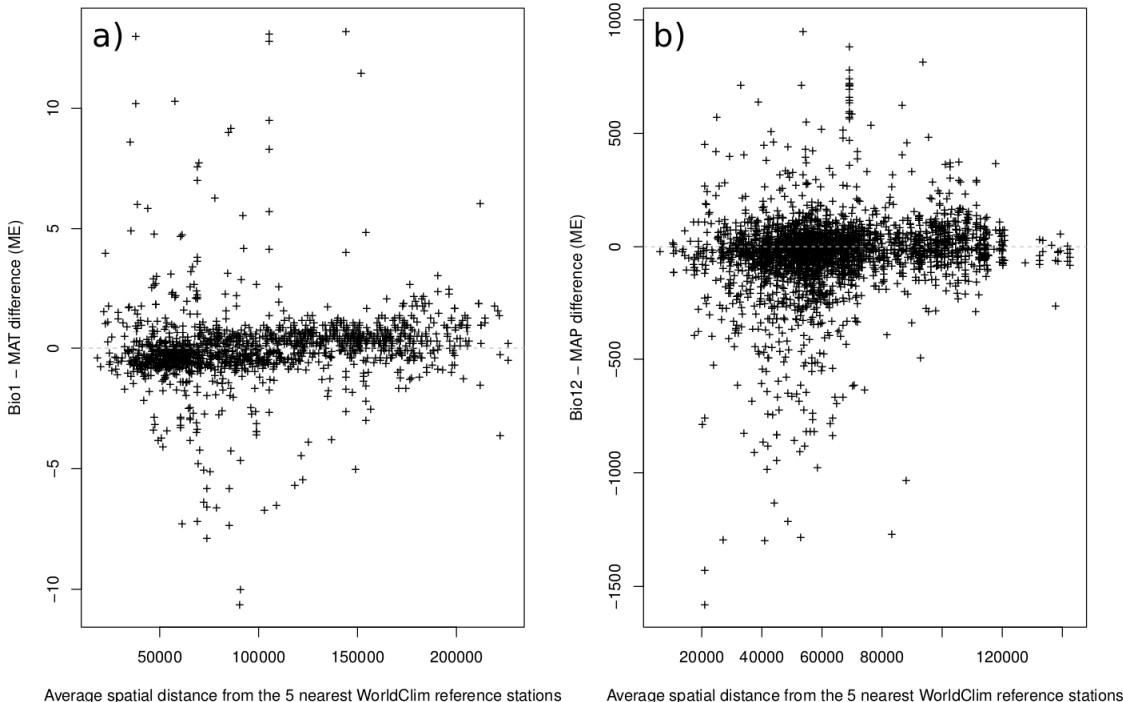

**Figure 4.** Relationship between the detected differences (WorldClim value, observed) and the average spatial distance of the observed record (new database) from the five nearest WorldClim reference stations for (**a**) temperature and (**b**) precipitation.

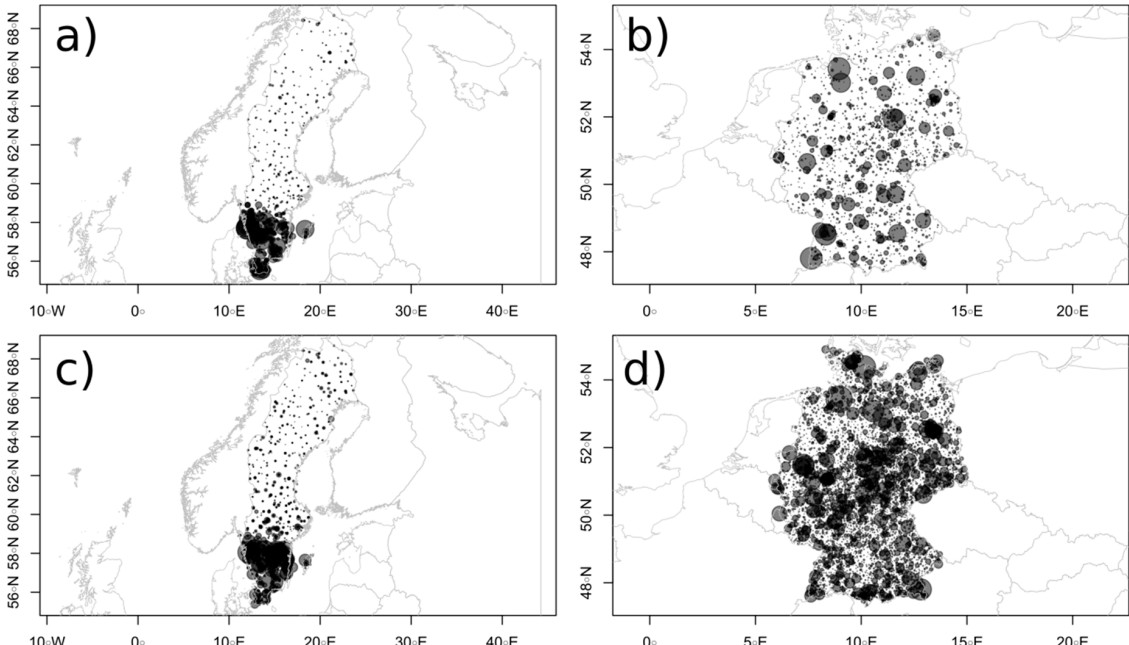

**Figure 5.** Spatial distribution of ABIAS for temperature (**a** and **b**) and precipitation (**c** and **d**) across Sweden and Germany, the two most sampled countries in the database. The larger the gray dot, the greater the difference between WorldClim and the independent dataset.

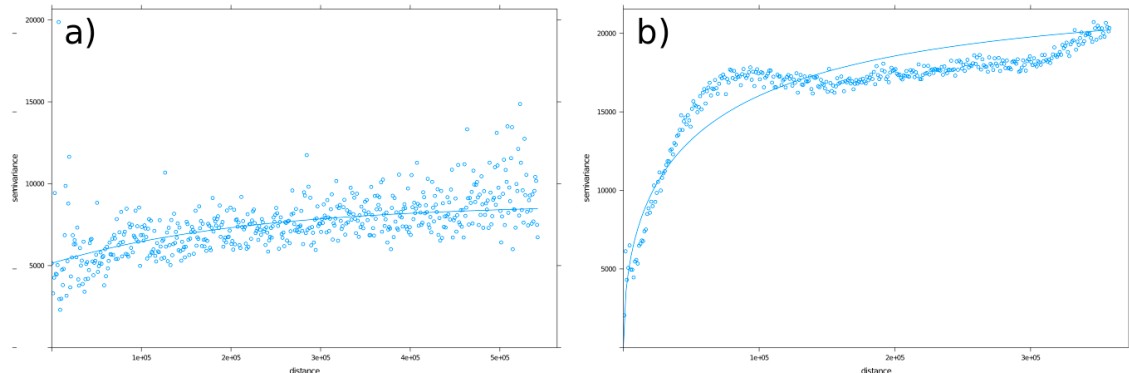

**Figure 6.** (Semi)variograms for precipitation calculated by the gstat package in R for Sweden (**a**) and Germany (**b**). There is a rather clear spatial structure that might be used as input for further geostatistical procedures (i.e., kriging) in order to adjust WorldClim raster maps.

## 4. Discussion

The quality of the reference baseline climate has a fundamental role in predictions of the potential impact of climate change on organisms and natural ecosystems. The stability and reliability of the estimated projections calculated from species distribution models [14,45,46], management simulators [18,47], or the estimation of the geographical shift of climate zones [48] rely on the differences between current and future climate. While the representativeness of WorldClim is adequate concerning air temperatures, large differences were found in the precipitation surfaces. Our results demonstrate a systematic difference of 0.76 °C between observed and interpolated values. According to the most recent IPCC report [49], the observed increase in temperature has been around 0.2 °C per decade. As a consequence such difference might affect future projections of WorldClim dataset adding uncertainties on the further modelling efforts [22,50]. In this case a likelihood analysis should be more adequate than deterministic ones and in order to include a sensitivity analysis and evaluate the probability of success of empirical models based on phenotypic plasticity and applied future projections [51]. Precipitation, by contrast, proved to be the main weakness of WorldClim surfaces in Europe. Despite its relatively small spatial extent, the European environment is characterized by many different forest systems that reflect broad climate variability, spanning from the Mediterranean to the Arctic.

The 1961–1990 baseline period is a fundamental dataset for ecological modelling because records from earlier periods were often affected by different instrumentation or changes in observational practice [30,37]. Therefore, numerous studies from climatology to biology, ecology and forestry [36,48,52] have used this baseline period, and WorldClim has been used extensively. We can expect a further warming trend in the next two decades at a rate of about 0.1 °C per decade, due mainly to the slow response of the oceans. As a consequence, even though the linear regression analysis showed a good match between observed and interpolated data (adjusted $R^2$ = 0.856), the difference is higher than the expected rate of change, which could heavily affect model predictions, adding uncertainties on future projections and smoothing results (i.e., land suitability projections) in an uncontrolled way [14,53–55]. This issue is then amplified when analysing MAP, where higher differences were found in combination with a poor regression analysis result. As a consequence, important biases may be introduced when using WorldClim's precipitation dataset. This is particularly true when WorldClim is used as the reference line and climate projections are locally downscaled and added to the WorldClim surfaces, as in the "Delta method" [56]. As a result, the calculation of climate indices might be difficult. For example, many studies used reference evapotranspiration [57–59] as the main predictor in statistical models [3,60,61]. In this case, the mathematical combination of differences in MAT and MAP might introduce uncontrolled biases through the study area. These biases could

represent a critical issue, especially in the Mediterranean and anywhere else that moisture deficit is identified as the most relevant climate driver.

The main advantage of our compiled dataset might be its representativeness at small scale. The authors of WorldClim themselves warn that the high resolution of the climate surfaces does not imply high data quality in all places as this depends on local climate variability, quality and density of observations and the degree of the fitted spline [31]. In a similar study, when compared with PRISM and Daymet datasets for the continental United States, many concerns were expressed, especially regarding the quality of WorldClim's precipitation grids in mountainous areas [34,35,62,63]. For this reason, several studies at regional or national scale at higher resolution (e.g., 100–250 m) preferred the use of meteorological variables obtained at nearby observational sites [28,64–66]. Regardless of the distances of the investigation sites from the locations where meteorological datasets were gathered, orography and land use, and the surrounding area and variable characteristics, must be considered. At small scale their variability may be a strong driver of frequently overlooked heterogeneities, leading to significant discrepancies in transferred datasets used for otherwise appropriate processing methods [67,68].

The lack of any relationship between BIAS and the main physiographic parameters (i.e., latitude, longitude, and elevation) does not allow for any statistical adjustment (e.g., downscaling, locally calibrated lapse rate, etc.) for either temperature or precipitation. However precipitation regimes are very difficult for meteorological stations to record properly and this issue has often been found in other databases [59,69]. Many more data are required, especially in the case of forest monitoring, as a result of the lack of temporal autocorrelation during the timeframe [70].

The need for a freely available and representative global climate dataset is large and growing, as evidenced by WorldClim's citation statistics. These goals can be achieved with local up-to-date monitoring networks, which could play a key role in evaluating global grids at small scale [71] as well as providing data for the construction of additional global climate datasets. Harmonization efforts, as well as increased representativeness of the established networks, are paramount for construction of more accurate climate surfaces. Enhanced data recovery with regular spatial coverage may overcome the lack of dense environmental or climatological sampling [28,70,72,73]. Derived surfaces are fundamental in order to plan future management strategies. For instance, and concerning forestry, additional strata, such as homogeneous climate zones, are needed as a fundamental tool to plan the transfer of genetic resources and reproductive materials across specific geographic areas [74,75]. WorldClim grids were interpolated with spline functions, a fast method known to yield results similar to polynomial functions but without mathematical instability. Such methods do not consider the spatial autocorrelation between observations, only partially achieved by more complex models where latitude and longitude are included as predictive variables [28,76]. Therefore, the exhibited spatial aggregation of the BIAS in the case of denser observations of our dataset (i.e., Sweden and Germany) may be relevant for research activities and improvements of the climate surfaces.

## 5. Conclusions

A new updated beta version of WorldClim has recently been released for the 1971–2000 time period. This "Version 2" (http://worldclim.org/version2), along with the need for carefully evaluating the quality of records used for modelling and keeping climate databases up-to-date, is an essential requirement for the adequate development of tools and informative systems. The lack of reliability on MAP values can be seen as the main shortcoming of the WorldClim database in Europe and elsewhere. However, precipitation is much more difficult to interpolate, given its low spatial and temporal autocorrelation as well as the lack of statistical relationships with some of the main physiographic parameters, such as elevation. Further research should focus on this parameter, seeking more significant determinants of MAP, given its importance in climate change scenarios where drought stresses are predicted to be the most relevant issue.

**Author Contributions:** Conceptualization, M.W., M.B. and M.M.; methodology, M.M. and M.W.; software, M.M.; validation, M.M., I.S. and M.W.; formal analysis, M.M.; investigation, M.M., M.B. and M.W.; resources, M.B.; data

curation, M.M., I.S. and M.W.; writing—original draft preparation, M.M. and M.W.; writing—review and editing, M.M., I.S., M.B. and M.W.; visualization, M.M.; supervision, M.B. and M.W.; project administration, M.W. and M.B.; funding acquisition, M.M., M.W. and M.B.

**Funding:** Maurizio Marchi was funded by EU, in the framework of the Horizon 2020 B4EST project "*Adaptive BREEDING for productive, sustainable and resilient FORESTs under climate change*", UE Grant Agreement 773383 (http://b4est.eu/). Marjana Westergren and Iztok Sinjur were funded by the Slovenian Research Agency, Research programme Forest Biology, Ecology and Technology P4-0107.

**Acknowledgments:** We thank Gregor Vertačnik from National Meteorological Service of Slovenia and Athanasios D. Sarantopoulos from the Hellenic National Meteorological Service Division of Climatology for contributing data.

**Conflicts of Interest:** The authors declare no conflict of interest.

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
