# Peer review of "Evaluating WorldClim Version 1 (1961–1990) as the Baseline for Sustainable Use of Forest and Environmental Resources in a Changing Climate"

_sustainability, doi:10.3390/su11113043_

Round 1

Reviewer 1 Report

The manuscript presented for review is a methodological study. It is interesting and raises current problems. However, before publishing, manuscript requires corrections.

The notes to the text are detailed below:

1) Introduction: it seems that the data presented is important not only for forests, but also for other ecosystems (lines 1-33). I understand that the authors focus on forests, but other ecosystems should also be mentioned.

2) For what purpose are the Table and Figure subtitles in the text highlighted in yellow?

3) Tab.1. - in the column "data source" sometimes there are hyperlinks => should be everywhere, or not at all, should be standardized.

4) Fig.1. - add the letters a and b (according to the instructions for the authors of the journal) and the appropriate markings in the title of the chart. There is no explanation regarding the differences in the density of dots in the graphs. What do the dots mean? what do they relate to? what scope are they presenting?

5) Line 139 - should be Table, space, 2, period

6) Table 2. - the abbreviation for "standard deviation" is "std", not "st.dev."

7) Tab.3. - shortcuts can be used instead of full names

8) Fig. 2; 3; 4; 5 - add the letters a and b (according to the instructions for the authors of the journal) and the appropriate markings in the title of the chart.

Figures are not self-explanatory, it is necessary to expand the description of data presented in figures in the text of the manuscript.

9) line 272 - what does the expression "ans" mean?

Author Response

Rev#1

The manuscript presented for review is a methodological study. It is interesting and raises current problems. However, before publishing, manuscript requires corrections.

The notes to the text are detailed below:

1) Introduction: it seems that the data presented is important not only for forests, but also for other ecosystems (lines 1-33). I understand that the authors focus on forests, but other ecosystems should also be mentioned.

Authors: some parts in the introduction and in the discussion section have been add to include this aspect in the paper. Please note that also the title has changed to deal with this

2) For what purpose are the Table and Figure subtitles in the text highlighted in yellow?

Authors: It was an editing mistake. Actually we use this method to control the figures and tables citations in the text. We were sure we removed it but we don’t. Now all the highlighted text has been removed

3) Tab.1. - in the column "data source" sometimes there are hyperlinks => should be everywhere, or not at all, should be standardized.

Authors: changed as requested. Now all the data sources are hyperlinks. Again it was an editing problem given by the use of unix-like systems (Ubuntu) and then an upload in a .docx format

4) Fig.1. - add the letters a and b (according to the instructions for the authors of the journal) and the appropriate markings in the title of the chart. There is no explanation regarding the differences in the density of dots in the graphs. What do the dots mean? what do they relate to? what scope are they presenting?

Authors: a wider explanation of the figure was add in the figure’s caption following also the comment number 8. We believe that now the figure is more readable

5) Line 139 - should be Table, space, 2, period

Authors: a typo error. We apologize

6) Table 2. - the abbreviation for "standard deviation" is "std", not "st.dev."

Authors: changed, thanks

7) Tab.3. - shortcuts can be used instead of full names

Authors: column names were shortened using shortcuts

8) Fig. 2; 3; 4; 5 - add the letters a and b (according to the instructions for the authors of the journal) and the appropriate markings in the title of the chart.

Authors: all the figures were adjusted following the suggestion

Figures are not self-explanatory, it is necessary to expand the description of data presented in figures in the text of the manuscript.

Authors: the caption of all the figures was enriched in order to solve this shortcoming

9) line 272 - what does the expression "ans" mean?

Authors: a typo error. The correct word is “and”. We apologize

Reviewer 2 Report

It is an interesting paper but it needs strong improvements. See file attached.

Author Response

L10-24: Highlight the findings and the their added value. It is better for a paper if a reader is able to cite it only by reading the abstract. You should make it understandable even for multidisciplinary redearship.

Authors: the abstract has been enriched with more details on the results as requested

L20: predictors the Author used → predictors used

Authors: changed, thanks

L35: and forest → and forest

Authors: the sentence has changed

L51-52: biog- georaphical → bio- georaphical

Authors: changed, thanks

L79-80: the discrepancies from the WorldClim climate surfaces with measured data in Europe Unclear syntax- restructure

Authors: we reshaped the whole paragraph to make it more readable

L83: an evaluation of the quality of the climatic surfaces Clarify the term “climatic surfaces” so as to be clear even for multidisciplinary readership. Clarify also what “quality” of them means. Is there “bad” or “good” climatic surfaces? In what sense?

Authors: in general the term “surface” is often used as synonym of “GIS raster map”. However we reshaped the whole paragraph to make it more readable

L141-142: 266 ± 988.64 is unclear

Authors: the ± 988.64 is the standard deviation in the number of records per country. We simplified the sentence to make it more readable

L154-157: ...(adjusted R2= 0.856) ...(adjusted R2= 0.642) … Why there is such a gap between R2 of MAT and R2 of MAP? Can you discuss it? This would be a useful basis for a question for future

research in the conclusion. I mean, the much lower R2 in MAP model implies that there are many more determinants which have not been measured while the MAT model much complete. In other words, the MAT is much more accurately predictable than MAP and, thus, the precipitation is more

complicated phenomenon than temperature. If you agree with this, the future research should be focused on MAP in order to find out more significant determinants of MAP and to increase the R2.

Authors: thanks for the useful suggestion. We add this short part in the conclusions as suggested

L158: A s consequence, Consequently,

Authors: changed, thanks

L159: degree of matching between the two regression lines was detected. Give a short clarification here in parenthesis about what these lines mean

Authors: The aim of the sentence is to demonstrate that Worldclim is charachetized by lower values before 500 mm and by higher values than those observed on the meteorological stations after this threshold.

L166: Bio1, bio2 Give a short clarification here in parenthesis about what these bio1 and bio2 mean

(what are the independent variables of the models discussed in 154-157 lines?

Authors: the caption of the figure was enriched to make it clearer

L218-222: ...a rate of about 0.1°C per decade, due mainly to the slow response of the oceans. As a consequence, even if the linear regression analysis showed a good matching between observed and

interpolated data (adjusted R2= 0.856) such difference is higher than the expected changing rate which could heavily affect model predictions. This issue is then amplified when analysing MAP where higher differences were found in combination with a scarce regression analysis result. This is a focal point. It is not quite clear and needs further clarification.

Authors: we tried to sold this issue clarifying the text and adding some references on that

L227-228: In this case the mathematical combination of the differences in MAT and MAP might introduce uncontrolled biases Unclear and important. It needs clarification. How have you ascertained biases and what kind of biases?

Authors: as for the question above we tried to clarify this properly

L258-260: Enhanced data recovery with a regular spatial coverage may overcome the lack of dense

environmental/climatological sampling. Too trivial (or not?) Do you have anything to say about the possible role of stations scarcity in biases? A spatially scarce sampling is necessarily a biased sampling?

Authors: some references on spatial interpolation methods and modelling effort, as well as scarcity of raw meteorological data were add to support this

L272 ans ?

Authors: a typo error. We apologize

L271-280:  Too technical and journalist passage. Try to restructure the conclusion highlighting concisely your main findings and their added value. I suggest to add also questions for future research

Authors: the conclusion section was totally reshaped

Reviewer 3 Report

The analysis of the accuracy of databases is an interessant and relevant issue.  The paper has value, however there are some issues to be dealt with:

Introduction-It’s OK. Although the authors, may exemplify more previous studies regarding the use of the database.

L 104-“Only very few points, corresponding to less than 1% of all the records, were lying outside country borders or land mass due to coordinate uncertainties, a very important aspect reflecting the high-quality of the new database.”-Exactly how many points and in which areas?

New version of worldclim is available, “using has average monthly climate data for minimum, mean, and maximum temperature and for precipitation for 1970-2000.” You could consider validating the data using this version.

Materials and methods-I recommend presenting a graphical scheme of the approach.

The statistical approach, although simple is reliable.

Results-The results are satisfactorily presented.

Discussion-It will well presented

Conclusion-The main conclusions of the work should be mentioned. The conclusion is quite short and syntetic.

Author Response

The analysis of the accuracy of databases is an interessant and relevant issue.  The paper has value, however there are some issues to be dealt with:

Authors: thanks

Introduction-It’s OK. Although the authors, may exemplify more previous studies regarding the use of the database.

Authors: more examples were provided

L 104-“Only very few points, corresponding to less than 1% of all the records, were lying outside country borders or land mass due to coordinate uncertainties, a very important aspect reflecting the high-quality of the new database.”-Exactly how many points and in which areas?

Authors: add in the text

New version of worldclim is available, “using has average monthly climate data for minimum, mean, and maximum temperature and for precipitation for 1970-2000.” You could consider validating the data using this version.

Authors: Yes but not in this paper whose aim is to evaluate the 1960-1990 period only as a pre-industrial baseline

Materials and methods-I recommend presenting a graphical scheme of the approach.

Authors: we welcome the recommendation and a new scheme has been created and included as Figure 1 in the manuscript. Now the total figures are 6

The statistical approach, although simple is reliable.

Authors: thanks

Results-The results are satisfactorily presented.

Authors: thanks

Discussion-It will well presented

Authors: thanks

Conclusion-The main conclusions of the work should be mentioned. The conclusion is quite short and synthetic.

Authors: this par has been partially rewritten in order to deal with Rev#1 and Rev#2 comments

Reviewer 4 Report

This is a good and most relevant paper. While it is somewhat out of my expertise, as much as I can understand it the study that is the basis of the paper is solid. The text is highly technical and authors should as much as possible try to write the paper such that it becomes accessible to a larger number of readers. The English, however, is quite poor and needs serious improvement.

Author Response

This is a good and most relevant paper. While it is somewhat out of my expertise, as much as I can understand it the study that is the basis of the paper is solid. The text is highly technical and authors should as much as possible try to write the paper such that it becomes accessible to a larger number of readers. The English, however, is quite poor and needs serious improvement

Authors: Thanks for your words. We worked to improve the grammar and English style with the help of our colleague who kindly modified our paper without asking to be add as Author. We hope this new version might be more readable and pleasant and, in the end, suitable for publication in sustainability journal

Round 2

Reviewer 2 Report

Excellent